# Chemical space docking enables large-scale structure-based virtual screening to discover ROCK1 kinase inhibitors

Paul Beroza [1] ✉, James J. Crawford [1], Oleg Ganichkin[2], Leo Gendelev[1], Seth F. Harris[3], Raphael Klein[4], Anh Miu[5], Stefan Steinbacher [2], Franca-Maria Klingler [4,6] & Christian Lemmen[4]

With the ever-increasing number of synthesis-on-demand compounds for drug lead discovery, there is a great need for efficient search technologies. We present the successful application of a virtual screening method that combines two advances: (1) it avoids full library enumeration (2) products are evaluated by molecular docking, leveraging protein structural information. Crucially, these advances enable a structure-based technique that can efficiently explore libraries with billions of molecules and beyond. We apply this method to identify inhibitors of ROCK1 from almost one billion commercially available compounds. Out of 69 purchased compounds, 27 (39%) have $K_i$ values < 10 μM. X-ray structures of two leads confirm their docked poses. This approach to docking scales roughly with the number of reagents that span a chemical space and is therefore multiple orders of magnitude faster than traditional docking.

Virtual screening aims to computationally search the universe of potential organic molecules to identify a manageable number of virtual "hits", whose physical samples can be obtained and tested in the laboratory to assess their activity on a desired target. One such computational method, molecular docking, is widely used in drug discovery initiatives. It uses the three-dimensional structure of the target protein and places small molecules into its binding site. There is, however, a computational complexity inherent in the docking process: the evaluation of each small molecule requires consideration of its various low-energy three-dimensional conformations, each of which has a distinct 3D geometry. Further, a larger screening library should improve both the number and quality of the hits identified[1], thereby increasing the downstream chances of drug discovery success. As a result, ever larger libraries of compounds have been considered, with recent docking campaigns reaching the billion-compound milestone, but only through the deployment of significant computational resources[2,3].

The development of high-throughput docking has been accompanied by a significant growth in the number of unique small molecules that can be obtained. Most significantly, the availability of "synthesis-on-demand" molecules, in which well-validated synthetic routes are coupled with ever-expanding lists of building blocks, has led to a combinatorial explosion in the number of molecules available for purchase. The chemical supplier Enamine's REAL Space, for example, comprises more than 20 billion compounds (as of 07/2021). While this seems large, it is dwarfed by the numbers obtained through combinatorial synthesis based on published reaction protocols: KnowledgeSpace[4] with $10^{14}$ compounds or that reported by GlaxoSmithKline with $10^{26}$ molecules[5].

Docking of such large combinatorial spaces is impossible with current hardware limitations. Even enumeration of the products is not feasible, so potential products can only be stored as building blocks (see Methods for further description of building blocks) and connection rules that allow for on-the-fly compound generation. One approach to explore these large chemical spaces uses a similarity method called Feature Trees[6,7]. The algorithm compares a query molecule to all building blocks, and the best building blocks are joined with complementary building blocks to make new molecules,

[1]Discovery Chemistry, Genentech, South San Francisco, USA. [2]Proteros Biostructures GmbH, Planegg, Germany. [3]Structural Biology, Genentech, South San Francisco, USA. [4]BioSolveIT GmbH, St. Augustin, Germany. [5]Biochemical and Cellular Pharmacology, Genentech, South San Francisco, USA. [6]Present address: MSD, London, England. ✉e-mail: berozap@gene.com

according to the chemistry rules that define the space. This approach has been used successfully numerous times in virtual screening of very large chemical spaces[8].

The first approach to explore chemical spaces in three dimensions in a similar combinatorial manner, without brute-force enumeration, was FlexNovo[9]. It placed chemical fragments in an active site and iteratively expanded the most promising candidates by joining them with other fragments. However, the chemical spaces explored were based on splitting and recombining existing molecules[10,11], without consideration of reaction rules. This inherently limited the synthetic feasibility of the chemical space that could be explored, since only a subset of fragment combinations could actually participate in validated chemical reactions. In the current work, we restrict the combination of fragments to those that can participate in validated chemical reactions, allowing us to navigate the same chemical space that is defined by the synthesis-on-demand virtual libraries of reactions and their associated building blocks. Through the combination of structure-based evaluation of fragments with the reaction rules from established chemical spaces, we can explore larger chemical spaces that were previously restricted to 2D search methods. Our strategy, which we term Chemical Space Docking, capitalizes on the synergy of the exploration of large chemical spaces and easy access to physical samples of identified hits. This approach can be considered an extension of earlier work on combinatorial docking, in which a known binding moiety was expanded into products using a single reaction scheme[12–14].

## Results

### Selection of building blocks by docking

Building block fragments of the two-component chemical space (136,835 in total derived from the 71,894 building blocks (see Methods): a building block may have moieties that can participate in different reactions and therefore give rise to more than one building block fragment) were docked with the FlexX docking application[15] into the binding site of 2ETR with pharmacophore constraints (see Methods). Up to 10 docked poses per fragment were generated, which led to a total of 129,125 poses. These were assessed with the HYDE scoring function[16,17], and the top-scoring 50,000 poses were imported and inspected in SeeSAR[18] to select the best 500 unique building block

fragments and associated poses. Selection criteria for the initial docked poses were:

- Additional hydrogen bond interaction: along with the required hinge-binding pharmacophore, at least one additional hydrogen bond was required between ligand and protein.
- Ligand efficiency: the molecular weight of the fragments was required to range from 45 to 450 g/mol. Preference was given to small fragments with high docking scores.
- cLogP: fragments with calculated LogP values over 4 were excluded.
- Linker geometry: the docked pose of the fragment was required to orient its reactive moiety in a geometry that would result in a product with a potentially favorable interaction with the protein. Vectors that pointed toward the protein interior or toward the solvent were excluded.
- Torsion energy: docked fragments with high torsion energies were excluded[19,20].
- Chemical diversity: preference was given to chemically diverse and interesting scaffolds.

Representative fragment poses are shown in Fig. 1. The pose orientations and reaction vectors span the binding site and provide good coverage of its volume by the enumerated complete combinatorial products.

### Expansion of libraries and filtering of products

Following the chemistry rules of the two-component chemical reactions, each of the selected 500 docked fragments was used as the starting point for a full library enumeration of products[21]. The resulting 500 libraries yielded a total of 5,236,824 products. Each of the 500 libraries was docked using the previously computed fragment pose as a template with FlexX. For each product, up to five docked poses were generated, resulting in 23,305,389 docked poses of complete virtual products. These were subsequently scored with the HYDE algorithm, and the relaxation, clash-removal, and score-optimization during this step led to the rejection of over half of the poses for a total of 10,391,986 for further evaluation and filtering.

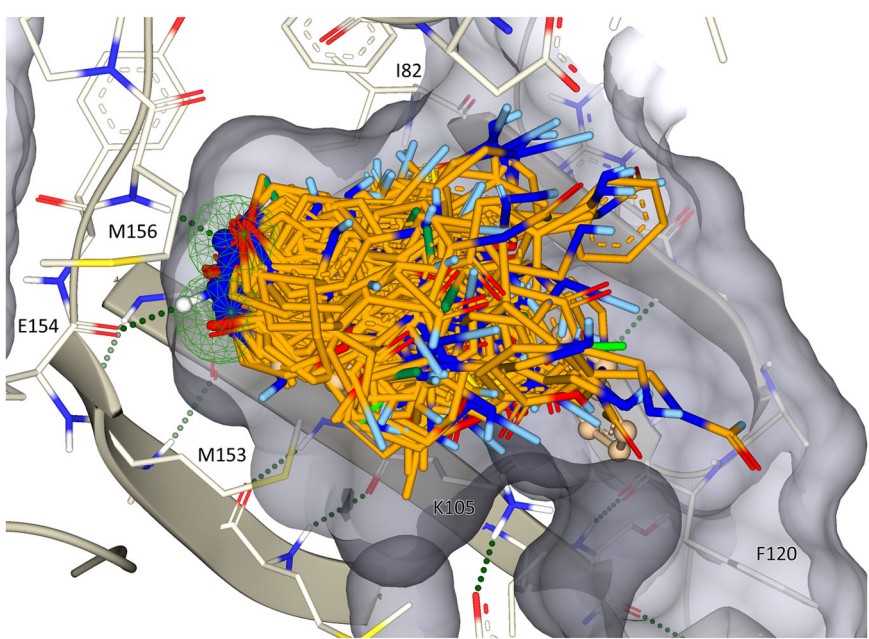

**Fig. 1 | Docked poses of selected starting fragment poses.** A representative set of docked poses (80 of the 500 initial fragments) are shown in the ROCK1 binding site. The binding pocket surface is shown in gray, and the hinge-binding pharmacophore is shown as two green spheres. Linkers are shown in light blue and serve as reaction vectors for product library enumeration (each fragment represents a library of ~10k molecules).

The top scoring 50,000 poses were selected and out of these the best poses per molecule (~33,000 virtual products) were chosen for further analysis. Strain energy filtering was done with the Chemalot software package[22]. Docked poses with an internal strain energy of five kcal/mol were removed, which reduced the number of molecules to 5940. To corroborate the docking results from FlexX, we redocked the remaining 5940 to the ROCK1 structure using the FRED docking program[23]. Further, in order to increase the chemical diversity of the compounds selected, we performed k-means clustering on the 5940 compounds using the default k-means clustering in Vortex[24], resulting in 500 clusters. From each cluster of compounds, the cluster member that had the best FRED docking score was selected for further evaluation.

The 500 cluster representatives were subjected to two qualitative filters: compounds with excessive flexibility and those containing large hydrophobic groups were rejected. Of the compounds remaining, a final set of 77 was chosen by visual inspection and ordered for purchase from Enamine (www.enamine.net). Of those, physical samples were obtained for 69.

## Analysis of active molecules

Of the molecules obtained, 27 had $K_i$ values below 10 μM (the upper limit of activity detection for the assay), corresponding to a hit rate of 39% (Enamine catalog IDs listed in Supplementary Information). The most potent compound was 38 nM, and 13 compounds (19%) had submicromolar potencies. Structures and ROCK1 $K_i$ values for the active compounds are shown in Supplementary Table 1. Many of the active molecules identified are structurally similar to known ROCK1 actives (see Supplementary Note 3 for analysis), which demonstrates that our method can deliver what is found through traditional screening and medicinal chemistry.

Figure 2 shows the initial fragment hits that led to the active molecules shown in Supplementary Table 1. The two phenylpyrazole fragments clearly yielded not only the largest number of actives but also the most potent ones. Beyond these two starting fragments, the others show an interesting structural diversity in their ability to interact productively with the kinase hinge region while optimally orienting a reactive moiety to have favorable interactions elsewhere in the binding site after the full product is docked.

The active compounds were grouped by their hinge-binding motifs: pyrazoles, lactam/pyridones, azaindoles, and indazoles. Each of the four groups consists of at least three active molecules. It is worth noting that the ligand in the structure used for the docking calculations contained a pyridine hinge-binding motif, and that motif was not represented in the active molecules identified. Figure 3 shows the binding pose of the most potent active molecule from each of the four chemotypes. In addition to the hinge-binding motif, which was required of all docked compounds, all active molecules identified in the virtual screening campaign contain a hydrophobic group that interacts with the P-loop of the kinase. This is particularly interesting because the ligand in the complex structure used for the docking calculations does not interact with the P-loop in this way; that volume in the protein/ligand complex is unoccupied.

The pyrazole class was the most populous, with fifteen active molecules. Perhaps unsurprisingly, these also featured the greatest structural diversity and the most potent examples. Examination of their docked poses revealed that they contained a phenyl group distal to the hydrogen bonding N-N pair of the pyrazole (Fig. 3a). This phenyl-pyrazole moiety fills a similar volume to that of the purine group in a native ATP-bound kinase structure. In addition to the phenyl group, all actives contained a C- or N-linked amide group at the *para* position to the pyrazole, providing the connection to the hydrophobic P-loop group.

The second most populous group, the lactam/pyridones, comprised three fused heterocycle hinge binding motifs: isoquinolinone,

dihydroisoquinolinone, and isoindolinone. For these compounds, both the carbonyl oxygen and the amide nitrogen form hydrogen bond interactions with the protein backbone of the hinge (Fig. 3b). As with the pyrazole inhibitors, the lactam/pyridones all have a hydrophobic group tucked under the P-loop. The saturated 6-membered example (compounds **18**, **19**, and **21**) is quite unprecedented for ROCK1 inhibitors. There were no examples of that motif as a hinge binder in the ChEMBL ROCK1 actives.

The three azaindole inhibitors have a distinctive tetrahydropyridine ring linking the putative hinge-binding group with the atoms that interact with the P-loop (Fig. 3c). This group also contains the most potent compound outside of the pyrazole hinge-binders.

Finally, the indazoles comprised three approximately equipotent actives whose docked poses bridge the hinge binding motif to the P-loop interaction by way of an acyclic amide. In two of the indazole analogs, a pyrazole group interacts with the P-loop and forms a hydrogen bond interaction with the catalytic lysine (Fig. 3d). In the third analog, the amide linkage is flipped, and the amide carbonyl group interacts with the catalytic lysine.

## Confirmation of docked poses by X-ray crystallography

To facilitate further validation of the method and confirm the postulated binding modes, we obtained co-crystal structures of ROCK1 in complex with Compound **1**, the most potent inhibitor identified in the virtual screening campaign, and with Compound **22**, an inhibitor we considered structurally unique. The structures were solved and refined to a final resolution of 2.34 Å (Compound **1**) and 2.74 Å (Compound **22**). The crystals contained four monomers of the ROCK1 kinase domain in the asymmetric unit, of which chains A, B, and C were relatively well defined, while the fourth copy of chain D was quite poorly ordered and not used in analyses (see Methods). Focusing on the chain A example in each structure, the resulting electron density shows a clear binding mode for each ligand (Figs. 4a and 5a). These were placed and refined by contributors to this work who were blinded to the docking pose results. A comparison between the experimental protein-ligand complexes and the docking poses obtained in the virtual screen is shown in Figs. 4b and 5b. The root-mean-square deviation between the docked pose and the X-ray structure was 0.97 Å (Compound **1**) and 2.30 Å (Compound **22**).

With the Compound **1** example, the pyrazole nitrogens that bind the hinge motif are 0.8 or 1.0 Å apart between the experimental structure and the docked model, while the alignment of the protein atom component of these hydrogen bonds show 0.4 or 0.6 Å separations. The mid-ligand phenyl linker ring has a distinct tilt in the experimental structure, but the overall ligand position remains co-located with the docked model (Fig. 4b). This is despite a large difference in the protein structures where our crystal structure model was built, interpreting density to indicate an inversion of the DFG motif (which also leads into a disordered activation loop) such that the inward location of the phenylalanine 217 sidechain creates a very different platform surface for the base of the ligand binding pocket. Similarly, the tip of the P loop of our structure is slightly lifted relative to the crystal structure used as a template for the docking (2ETR), allowing some differences in the ligand methoxy ethyl tail moiety. Since docking protocols generally do not account for such large conformational changes and protein dynamics, the consistency of the experimental and docked ligand poses in this example are all the more remarkable.

Analogously for Compound **22**, the azaindole nitrogen atoms involved in hydrogen bond interactions to the kinase hinge are only 0.4 or 1.0 Å displaced between the experimental and docked poses. In this case, the displacement between the two molecules becomes larger as one moves distally to the hinge towards the back portion of the pocket underneath the P loop. For example, the carbonyl group on the docked ligand that interacts with the tip of the conserved kinase N lobe

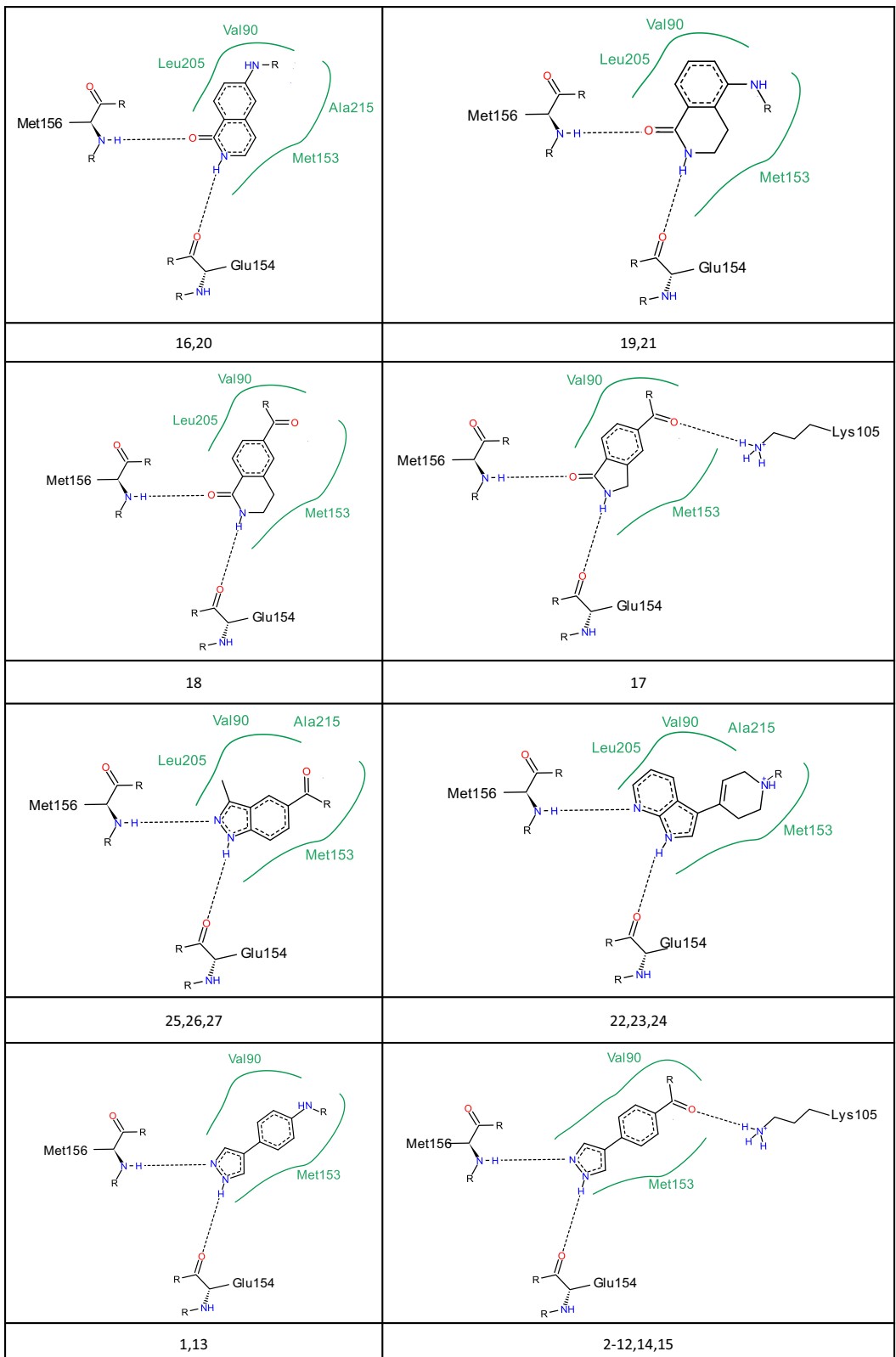

**Fig. 2 | Initial fragment hits.** Two-dimensional depictions of the initial fragment hits that led to one or more of the final 27 active molecule products. Key protein interactions with the ATP binding site are shown. Below each example is the list of active molecules in Supplementary Table 1 that were derived from the initial fragment hit.

lysine 105 is displaced 1.8 Å from the experimental structure and the respective distal phenyl groups underneath the P loop are 3.4 Å apart. However, it is critical to note that the protein itself demonstrates shifts of similar size in our alignment, such that we observe displacements

between our experimental structure and that used as a template in the docking procedure of 2.1 Å at the zeta nitrogen of Lys105 and 2.4 Å for equivalent backbone C-alphas (Phe87) at the P loop turn, or a 4.2 Å difference at the outer tip of that phenylalanine 87 sidechain. These

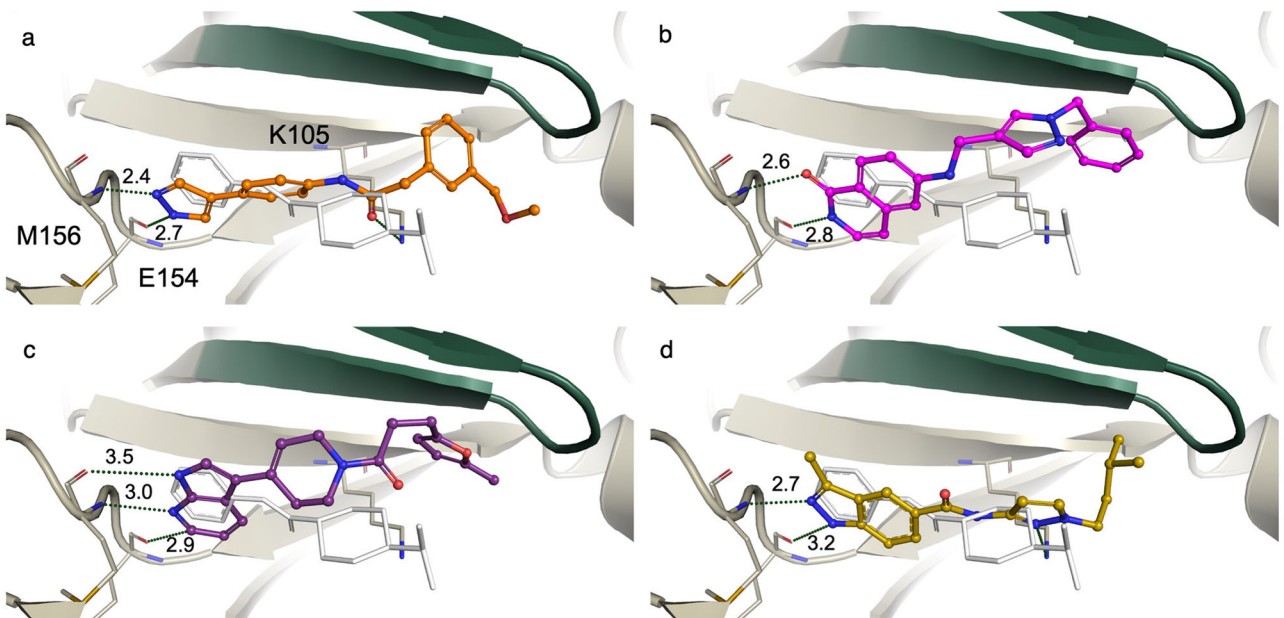

**Fig. 3 | Docked poses for the most active molecules from each of the four chemotypes identified.** The hinge residues are at the left and the P loop (green) is at the top of each panel, while the original 2ETR ligand is shown in thin white sticks for reference. **a** Compound **1** (pyrazole) in orange. **b** Compound **16** (pyridone) in magenta. **c** Compound **22** (azaindole) in purple d. Compound **25** (indazole) in yellow. Hits identified by Chemical Space Docking all interact with the kinase P-loop, and two interact with the catalytic lysine (see **a** and **d**). Neither of these interactions is present in the PDB ligand.

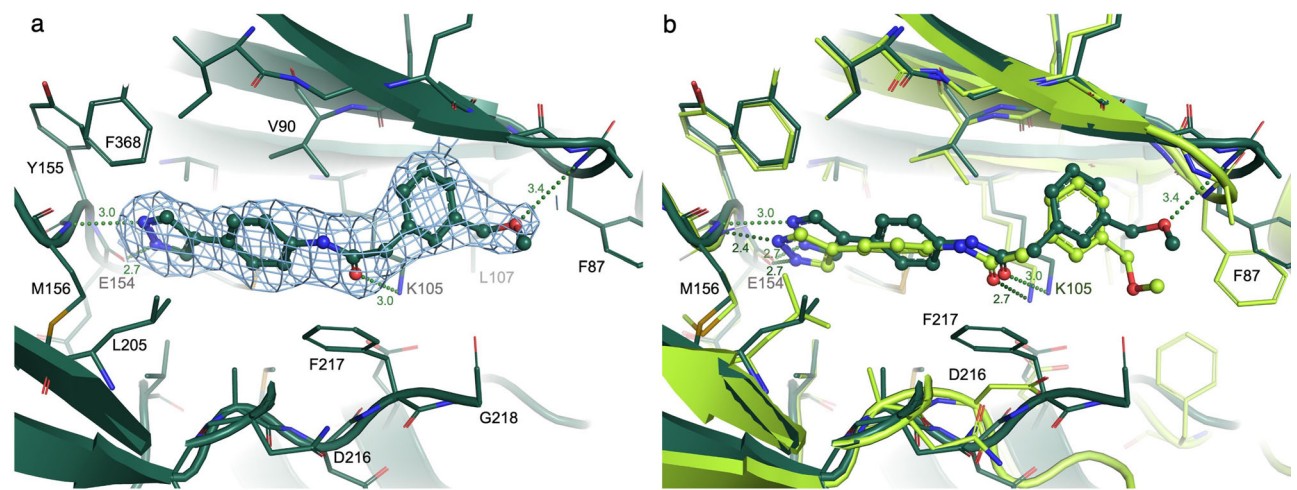

**Fig. 4 | Comparison of the X-ray structure of compound 1 with its docked pose. a** X-ray structure: the refined 2Fo-Fc electron density is depicted at 1 sigma contour in the vicinity of the ligand. **b** Overlay of the docked pose (light green) and X-ray conformation (dark green).

observations suggest that the bulk of the observed ligand displacement is due to the malleability and differences in the protein component that the docking algorithms would not be expected to achieve. For instance, ROCK1's helix C is partially unwound in the mid-section affording greater plasticity and this directly abuts the P loop and outer portions of the binding pocket. It is again reassuring that given those large adjustments in the protein, the ligand binding conformation itself is quite well matched between the two models, albeit with this tilted trajectory differential. Overall, these experimental structures confirm the expected binding modes.

During the execution of the virtual screening campaign, a publication reported a series of phenylpyrazole ROCK1 inhibitors, and a crystal structure of one of the inhibitors was deposited in the protein databank (PDB ID 7JOU)[25]. That structure has a very similar binding mode to the one we obtained, further validating our approach. The

second structurally characterized inhibitor, Compound **22**, and its analogs, contain a tetrahydropyridine linking group, which were not found in our survey of PDB kinase inhibitor structures. Comparison of the docked pose of the ligand with its X-ray coordinates shows good agreement with the ligand geometry in the hinge region, although the plane of the pyrrolopyridine ring is slightly rotated. This difference is magnified further from the hinge. There is some torsional variation in the terminal phenyl group, but there is good overall agreement between the two ligand geometries.

## Discussion

By typical metrics used to evaluate a screening campaign—hit rate, potency, and structural diversity of hits—this virtual screen succeeded beyond our expectations. A hit rate of 39% rivals the highest reported in the docking literature, and the potencies meet the criteria for most

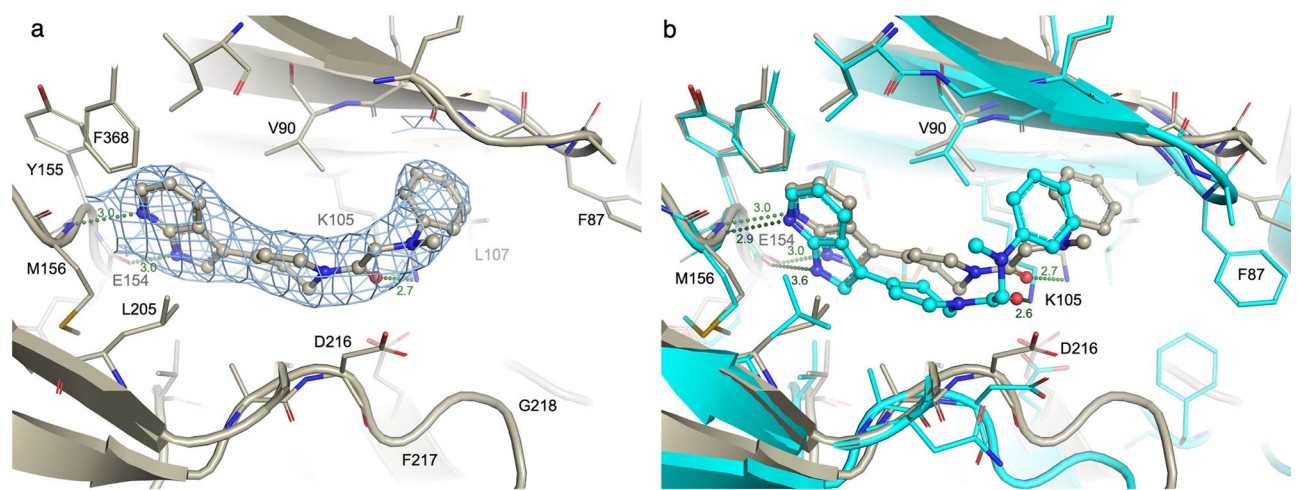

**Fig. 5 | Comparison of the X-ray structure of compound 22 with its docked pose. a** X-ray structure: the refined 2Fo-Fc electron density is depicted at 1 sigma contour in the vicinity of the ligand. **b** Overlay of the docked pose (cyan) and X-ray conformation (bone).

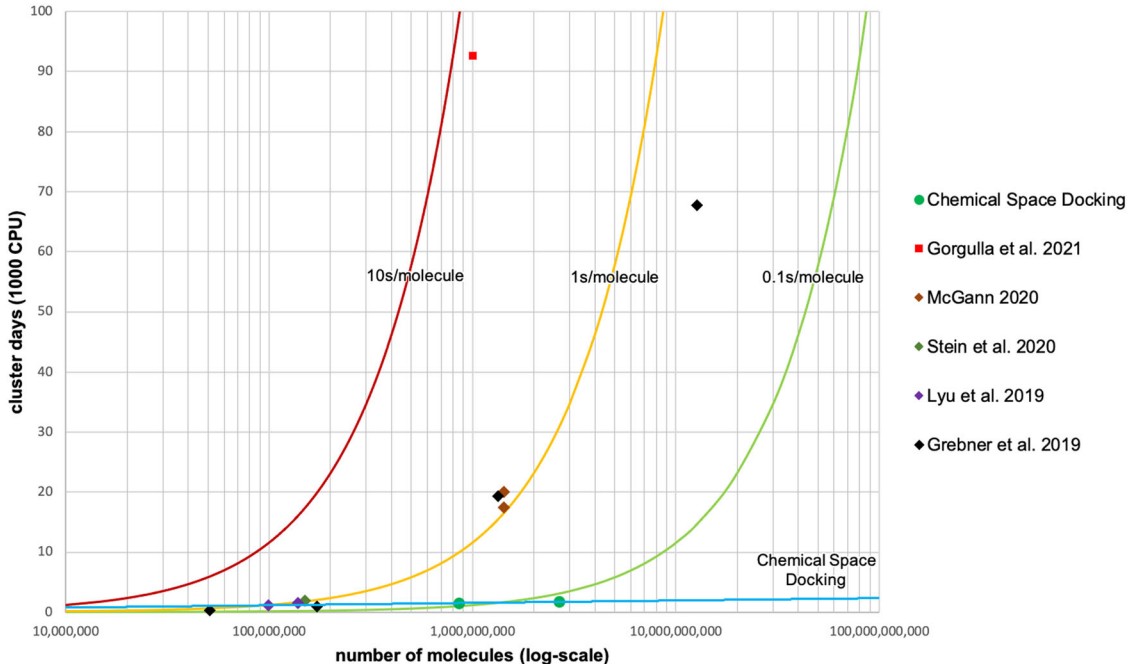

**Fig. 6 | Computational requirements for large-scale docking campaigns.** Traditional full enumeration docking curves are calculated based on the time needed to dock each molecule: 10 s (red), 1 s (yellow), 0.1 s (green). Chemical Space Docking curve shown in light blue. Large-scale docking campaigns from the literature are shown as individual data points (including Space Docking results described here and similar campaigns)[1,2,26,27,40].

biological screens for activity. Moreover, all of the hits have chemically related analogs that are also potent, which provides a nascent structure-activity relationship for each series. After the screening of only 69 compounds, the compounds identified could easily be starting points for a hit-to-lead medicinal chemistry effort on a therapeutic target in a modern drug discovery campaign.

The docking of reaction building block fragments and the selection of those that are most promising, followed by instantiation of the sub-libraries associated with them, provide an efficient, and in this example, highly successful alternative to current docking strategies based on enumerated libraries. Accordingly, our validation study opens the door to structure-based search of a much larger chemical space than was previously available. The ever-increasing reaction-based chemical spaces comprise numbers that are many orders of magnitude beyond what can be considered for docking completely

enumerated libraries. Even the computational requirements to create the products, much less to efficiently dock them, are daunting. Chemical Space Docking overcomes these obstacles.

The computational resources needed for different docking approaches are shown in Fig. 6. Recently published large-scale docking campaigns and their associated computation times[1,2,26,27] are compared with our docking approach. Traditional full library docking encounters resource limitations as the size of the chemical space increases. Our approach, with its much lower CPU requirements, is a much more efficient method.

One might argue that bigger and bigger libraries—let alone chemical spaces—are not needed[28]. It could be sufficient to work with a comparatively small, manageable, diverse library to find starting points for any drug discovery challenge. However, recent work has shown that a full virtual library screen is essential to identify the best

active compounds. Both the quality and quantity of hits deteriorates if smaller libraries are screened[1,29]. Another important recent finding is that even ultra-large chemical spaces are surprisingly unique[30], which makes the case for efficient search of large chemical spaces even more compelling. As long as the quality of the virtual screening hits is maintained, there seems to be good reason to cast the net as widely as possible in chemical space in the pursuit of bioactive molecules.

Although wide screening is preferable, computational resources are always limited. Our approach achieves a balance by focusing on the most promising building blocks for expansion, a so-called greedy optimization strategy. This has been applied quite successfully in numerous drug discovery applications, where complete searches are impractical[31]. Greedy methods require both additive scoring and optimal partial solutions to an optimal complete solution. To some extent this is a given here. An optimally scoring compound pose can be expected to score favorably in all of its component interactions within the binding site. Since the approach presented here starts with all building blocks considered in the first pass, chances are that for a multi-component, top-scoring molecule, at least one of its components will also be a top-scoring solution in the first pass. Further, the top-scoring component will likely survive as a solution through the greedy iterations. It should be noted that the practical restriction on the number of candidates that can be exhaustively searched by brute-force docking will probably pose a greater limitation to finding the best molecules than the unlikely omission introduced through the present greedy heuristic[29].

Comparisons between Chemical Space Docking and docking fully enumerated compound libraries are described in Supplementary Note 4. Enumeration of random selections from the full Enamine product space leads to a vast number of poor docking compounds, while Chemical Space Docking significantly enriches for good dockers. Comparison of Chemical Space Docking with complete enumeration of two select chemical subspaces shows a similar efficiency gain. While the majority of the best scoring compounds in the full enumeration were also found by our protocol, some high scoring compounds were not - the result of filtering the initial building blocks as described above. In any case, a significant fraction of high scoring compounds are found by Chemical Space Docking at a small fraction of the computational cost of brute force docking of an entirely enumerated compound space.

The docked poses of the starting fragments are critical for the success of the Space Docking approach. By design, candidate molecules are built from a single reaction vector for each candidate fragment pose. Ideally, the binding site provides guidance about where to initiate the building process: a deep pocket or a known required interaction. In the case of kinases, the hinge interaction, which is common to almost all orthosteric kinase inhibitors, provides such a docking anchor: the initial fragment has to make a hydrogen bond interaction with the hinge. Not surprisingly, the actives identified include chemical motifs that are known hinge binders[32,33]. Binding sites that do not possess an obvious point from which to grow molecules in situ might prove more challenging. In such cases, more poses for the initial fragments may be required to identify the best starting point.

Immediate plans to advance the results presented here include the automation of much of the workflow. For example, evaluation of the initial binding poses for the best geometry for further expansion was done largely by visual inspection; subsequent filtering of the enumerated products was also done sequentially by custom scripts and visual inspection. Many such steps may be automated with careful algorithmic development, which is currently underway. Further, extension of the method to three-component reactions will greatly increase the size of the chemical space considered in the search. Finally, kinases are well-established as druggable targets, and

application of the method to more challenging therapeutic targets is on-going.

The results presented here demonstrate that structure-based methods can be extended to the vast chemical spaces that were previously restricted to searches based on chemical graphs or reduced representations. Our hope is that informing such large-scale searches with protein structural information will greatly improve the number, quality, and novelty of chemical leads identified through virtual screening.

## Methods

### Docking protocol and chemical space definition
The initial stage of our virtual screening workflow was the definition of the appropriate docking protocol for the virtual screen. The 2ETR structure was chosen because its binding site could accommodate ligands from other PDB ROCK1 entries and had well-defined electron density in the ATP binding site (see Supplementary Note 1)[34]. Further, best docking results were obtained when the canonical kinase hinge-binding motifs were required: either (1) a ligand hydrogen bond donor was within hydrogen bonding distance of the backbone carbonyl of Glu 321 or (2) a ligand hydrogen bond acceptor was within hydrogen bonding distance of the backbone nitrogen of Met 323. The binding site and pharmacophore constraints used in the docking calculations are shown in Fig. 7. Further details of the calculations that led to the selection of the protein structure and the pharmacophore constraints are described in Supplementary Note 1.

### Building block fragments
The set of compounds to be searched with our reaction-based docking method was built from the two-component subset of the Enamine REAL Space compounds (reactions and building blocks were accessed in April 2019: https://enamine.net/compound-collections/real-compounds). This comprises 71,894 building blocks and 102 reactions, for a maximum possible total of 858,125,390 virtual product molecules. A building block is a molecule that contains a functional group that can participate in a chemical reaction. A building block fragment, or simply "fragment," is derived from a building block by retaining the part of the building block that remains after the reaction and introducing a dummy linker atom that defines the connectivity to a complementary fragment (i.e., the second building block in a two-component reaction). A building block may contain more than one functional group (compatible with different reactions) and can therefore give rise to more than one distinct fragment. All building block fragments were docked and the best poses were selected for further evaluation. Then, following the available reaction schemes, this set of anchor fragments underwent combinatorial expansion and docking of the complete chemical products. Finally, the docked products were further filtered to identify the most promising molecules.

### Biological assay
Human ROCK1 protein (amino acids 1–477 from accession number NP_05397.1) was purchased from Carna Biosciences (Cat# 01-109). This sequence contains the catalytic domain of the protein, and the biochemical activity was measured by the HTRF KinEASE-STK S2 Kit (Cisbio, Cat# 62ST2PEB) according to the manufacturer's protocol. To determine the $IC_{50}$, compounds were first dispensed by the Echo Liquid Handler (Labcyte) into white 384-well plates (PerkinElmer, Cat# 6008289). A total of 3 μL of 2x ROCK1 enzyme solution was added to the compounds, followed by a 10 min incubation at room temperature. Then, 2x ATP and STK S2 peptide solution were added to initiate the one-hour enzyme reaction at room temperature. The final condition of the reaction was 1.5 nM Rock1, 3 μM ATP, 0.5 μM STK S2 peptide in 50 mM HEPES pH7.2, 10 mM $MgCl_2$, 0.1% BGG, 0.005% Brij-35, 1 mM

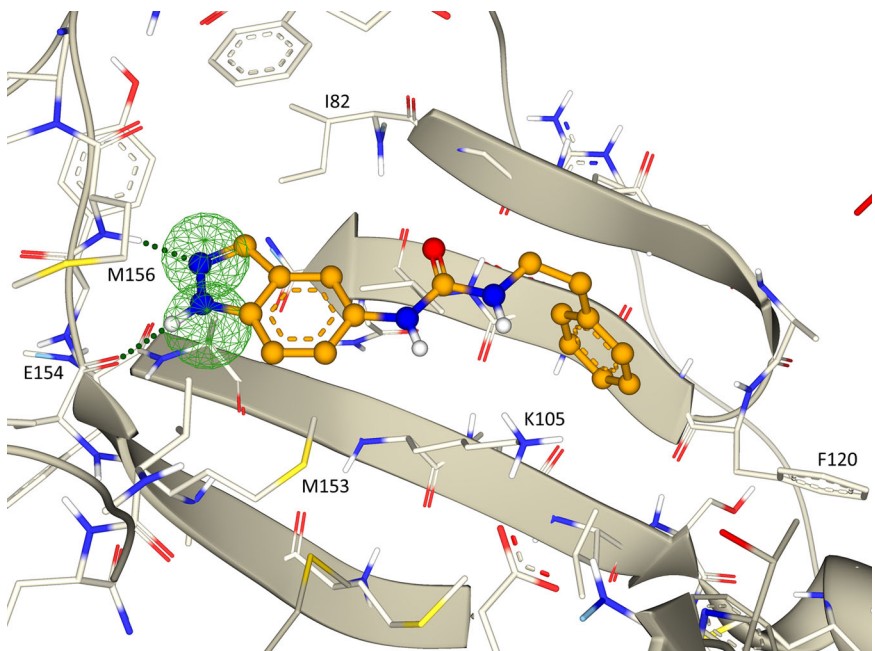

**Fig. 7 | The ATP binding site of ROCK1 (PDB ID: 2ETR).** The pharmacophore constraint for hydrogen bond interaction with the kinase hinge residues is shown as green spheres. A different kinase ligand (PDB ID 3V8S) is shown to illustrate both possible hinge interactions that are consistent with the pharmacophore constraint.

DTT. The reaction was quenched by adding 6 µL of the detection mixture that contained Streptavidin XL665 and STK Antibody-Cryptate (Cisbio) and incubated for 1 hour at room temperature. The HTRF (665 nm/620 nm) signal was read on the PHERAstar reader (BMG Labtech). All measurements were done in triplicate.

### X-ray structure determination

Human ROCK1 (kinase domain, residues 6 to 405, with N-terminal TEV-cleavable His tag) was expressed in insect cells. Purification was performed at 4 °C, with an initial Ni-NTA affinity chromatography step followed by TEV cleavage of the His tag. TEV-cleaved ROCK1 was passed over a second Ni-NTA column, and the ROCK1 protein from the flow-through was resolved by size exclusion chromatography in 100 mM NaCl, 20 mM HEPES/NaOH pH 7.5 and 2 mM beta-mercaptoethanol. Peak fractions were collected, concentrated to 22 mg/ml, flash frozen in liquid nitrogen, and stored at −80 °C. For crystallization the protein was diluted to 16 mg/ml with storage buffer. The protein was incubated for 1 h on ice with the respective ligand at 1 mM concentration (added from 100 mM DMSO stock). For crystallization (hanging drop) 0.5 µl of the protein/ligand solution, 0.5 µl of the reservoir solution were mixed and incubated over 100 µl of reservoir solution in Linbro Plate (Jena Bioscience GmbH). The reservoir solution contained 15–17% PEG5K MME, 0.1 M HEPES/NaOH pH 7.5, 5% (v/v) tacsimate pH 7.5. For cryoprotection with TMAO the Free Mounting System (FMS) was used[35]. X-ray diffraction data were collected at the Swiss Light Source (Villigen, Switzerland) on beamline PXII/X10SA with an EIGER detector (Dectris), and processed with autoPROC[36]. Molecular replacement was performed with MOLREP[37], model building with COOT[38], and refinement with REFMAC5[39]. Of the four copies of the kinase domain observed in the crystallography asymmetric unit, chain D had quite poor electron density suggesting significant disorder, yet sufficient signal to indicate its presence. We therefore excluded that molecule from analytical consideration in this work, focusing rather on the chain A example, where both the protein and ligand were relatively well defined. In an initial submission, side chain occupancies were lowered to zero for much of chain D; this was, however, deemed questionable by PDB validation and we have in response restored the occupancies to 1 throughout chain D despite the fact that some of the global and chain D local quality statistics suffer. While notably, the real space RSRZ fit criteria, higher B factors in chain D, and some elevation of the global R factors, nonetheless, chains A, B, and C provide more robust substrate for analysis as shown in the electron density figures throughout this work. The Ramachandran profiles show 0.07% outliers and 98.06% favored geometries for the ROCK1-compound 1 complex structure, and 0.22% outliers and 95.05% favored geometries for the ROCK1-compound 22 complex. See Supplementary Note 2 for further details.

### Reporting summary

Further information on research design is available in the Nature Research Reporting Summary linked to this article.

## Data availability

Source data are provided in this paper. Chemical structures and their analytical and biological characterization are presented in the manuscript and its Supplementary Information. Crystal structure coordinates and structure factors for compounds **1** and **22** are deposited in the PDB under accession codes 7S25 and 7S26. Source data are provided in this paper.

## Code availability

Software tools for docking and analysis:

FlexX 4.3 and SeeSAR 10.0: BioSolveIT GmbH, Sankt Augustin, Germany, www.biosolveit.de

FRED implemented in OEDocking 1.4.1: Open Eye Scientific Software (www.eyesopen.com)

MOE 2019.0104: Chemical Computing Group (www.chemcomp.com)

Vortex 2018.03.71496.53-s: Dotmatics, Inc. (www.dotmatics.com)

Chemalot cheminformatics code is available in Github: https://github.com/chemalot/chemalot (downloaded May 2020)

Software tools for X-ray structure determination:

COOT 0,9.6 and REFMAC5: 5.8.0258 MRC Laboratory of Molecular Biology (https://www2.mrc-lmb.cam.ac.uk/personal/pemsley/coot/)

MOLREP 11.0 (https://www.ccp4.ac.uk/).

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

## Acknowledgements

We thank Erik Evensen and Markus Lilienthal for technical assistance, Stephen Cox for editorial assistance, Sarah Robinson and Yong Wang for help with chemical characterization, and Hans Purkey, Aviv Regev, and Yiming Xu for helpful discussions. This work was funded by Genentech.

## Author contributions

P.B. and C.L. conceived the study. F.-M.K. performed the initial building block docking, subsequent library enumeration, and docking to identify 33,000 virtual products. P.B. performed subsequent filtering, crossdocking, and clustering. P.B., J.J.C., and L.G. made the final compound selections for synthesis. J.J.C. oversaw the chemical characterization of the synthesized molecules. A.M. performed the biological assays. O.G., S.S., and S.F.H. performed structural biology to obtain the X-ray

structures of two of the active compounds. R.K. conducted computational experiments to compare the method to the docking of fully enumerated libraries. All of the authors contributed to writing and editing the manuscript.

## Competing interests

P.B., J.J.C., L.G., S.F.H., and A.M. are employees of Genentech, Inc. R.K., F.-M.K., and C.L. are employees of BioSolveIT, GmbH. O.G. and S.S. are employees of Proteros Biostructures GmbH.
