## [Peer Review File · Nature Communications]

REVIEWER COMMENTS

Reviewer #1 (Remarks to the Author):

The manuscript deals with large-scale docking, targeting ROCK1 kinase and utilized fragment-based type docking approach. The manuscript is interesting and in general level results are encouraging, as several novel active compounds were identified. Also I think that produced X-ray structures are nicely underlining how well some of the kinase inhibitor binding poses can be predicted.

However, I have some questions and comments which, on my mind, has not been answered in the current version of the manuscript. The main point I have is the following: in the current form the manuscript is a good case study of hit finding for ROCK1 kinase. However, this study does not evaluate the performance of "Chemical Space Docking" approach. In the following I try to explain the main issues.

1. As I understand, the main benefit of the given virtual screening method is that full library enumeration and subsequent docking is not needed. Although current results are good, it is still not clear if those results are comparable to the case where full library is used (so how many hits are missed). I know this is a big task, but in the current form one cannot claim anything about the performance of the approach used. I am quite sure that the method SHOULD give good results, but now there is no information how large library must be docked in general to reach (for example) 80% hit rate if compared with full library docking. It is also unclear how big portion of the "full library" was actually docked. So this is more like an experiment with $n=1$ and without control.

2. The data is based on just one target (ROCK1 kinase). Like the authors also state, kinase proteins are well known targets and all active compounds are typically binding into the hinge-region. This, in principle means that virtual screening with pharmacophoric constraints will be very suitable for fragment-based methods (which Chemical Space Docking actually is). So the current study is based on searching small fragment-like compounds binding the hinge-region. If the target would be more general (for example nuclear receptor) I would guess that the method would be unable to identify as good compounds. In other words, does this method work only on those cases where we can easily identify optimal location for starting fragment?

3. It is quite unclear why three different scoring functions were used (FlexX, HYDE and FRED). And another (related question), how sensitive the approach is for used docking method. This is an important question, since different docking methods do utilize different ways for initial pose identification and thus small fragment-like docking is working differently.

4. Another main approach to solve the issue of large library docking/enumeration is to use machine learning (for example

Yang, Y. et al. Efficient Exploration of Chemical Space with Docking and Deep Learning. J Chem Theory Comput (2021) doi:10.1021/acs.jctc.1c00810.). If I compare the results from this manuscript to those of published Deep Docking studies, I don't see any major benefit here (speed or accuracy). How the authors would comment this aspect?

Sincerely Yours,

Prof. Antti Poso

Reviewer #2 (Remarks to the Author):

The authors present a well-written first successful approach to docking the contents of ultra-large fragment spaces. Large-scale fragment spaces are becoming more and more popular and structure-based methods to search them are highly desirable. So, the new method described is of high interest to the readers of your journal.

This is why I recommend the paper to be published after minor revision. Here are my points of concern:

- please check the grammar of the first sentence in the second paragraph of the introduction
- mentioning FlexNovo as the first approach in this context: what about combi-docking?
- please rephrase your differentiation to FlexNovo, as it is misleading in the current version:

the FlexNovo space is restricted to splitting and recombining products ignoring reaction rules

versus

the fragment space for docking is restricted to validated reactions

- please check:

In the results chapter you describe 136,835 building blocks compared to 71,894 building blocks in the methods section

- You got 129,125 poses in the first step. Do you know, from how many building blocks?

Or in other words, how many building blocks were declined by docking?

- In the first paragraph of the results chapter you selected the best 500 unique molecules. In my opinion, it is clearer to talk about the best 500 unique fragments or building blocks and the associated poses.

- You preselected the top scoring 33,000 virtual products. Is there any rationale for this number?

- Please correct the typo in the first sentence of the second paragraph of the results chapter.

- In the last paragraph of the results chapter I miss a reference or further details about compound 7. Obviously, it is not your hit no. 7 and where did you get the XRay structure from?

- Supporting information:

In the list of PDB codes are only 17 of 18 codes

- Supporting information:

which software and which Fingerprints did you use for the 2D similarity searches?

Response to Reviewers' Comments

Reviewer #1

1. *As I understand, the main benefit of the given virtual screening method is that full library enumeration and subsequent docking is not needed. Although current results are good, it is still not clear if those results are comparable to the case where full library is used (so how many hits are missed). I know this is a big task, but in the current form one cannot claim anything about the performance of the approach used. I am quite sure that the method SHOULD give good results, but now there is no information how large library must be docked in general to reach (for example) 80% hit rate if compared with full library docking. It is also unclear how big portion of the "full library" was actually docked. So this is more like a experiment with n=1 and without control.*

Comparison to a docking campaign on a full library is a big task, and we think it would require significant computational resources to execute. While our manuscript was under review, a similar greedy approach to docking very large libraries was published (Sadybekov, *et al. Nature* 601, 452–459, 2022). In that paper, for one of their validation exercises, they used the same ROCK1 pdb structure we used in our study. They compared their hit rate to a random selection of a full library and estimated a 40-fold enrichment. Our work corroborates theirs nicely and should give the scientific community confidence that the method is valid and useful.

2. *The data is based on just one target (ROCK1 kinase). Like the authors also state, kinase proteins are well known targets and all active compounds are typically binding into the hinge-region. This, in principle means that virtual screening with pharmacophoric constraints will be very suitable for fragment-based methods (which Chemical Space Docking actually is). So the current study is based on searching small fragment-like compounds binding the hinge-region. If the target would be more general (for example nuclear receptor) I would guess that the method would be unable to identify as good compounds. In other words, do this method work only on those cases where we can easily identify optimal location for starting fragment?*

This is an important point. Kinases are excellent candidates for this approach, because almost all orthosteric inhibitors have a hydrogen bond interaction with the hinge. Other proteins may be more challenging. As we mention in paragraph 6 in the Discussion: "Binding sites that do not possess an obvious point from which to grow molecules in situ might prove more challenging. In such cases, more poses for the initial fragments may be required to identify the best starting point." This is an area for our future work, but we feel confident that this building block approach can generalize as a docking approach that can greatly expand the chemical space considered in a docking campaign.

3. *It is quite unclear why three different scoring functions were used (FlexX, HYDE and FRED). And another (related question), how sensitive the approach is for used docking method. This is important question, since different docking methods do utilize different ways for initial pose identification and thus small fragment-like docking is working differently.*

The use of different scoring functions was a practical consideration. This effort was a collaboration, and one group used FlexX for docking (and the visualization that accompanies the HYDE scoring function), while the other used FRED for a docking engine. Interestingly, the work of Sadybekov, *et al.* used yet a different docking engine, and the results they got for the ROCK1 system are different from ours.

4. *Another main approach to solve the issue of large library docking/enumeration is to use machine learning (for example Yang, Y. *et al. Efficient Exploration of Chemical Space with Docking and Deep Learning. J Chem Theory Comput* (2021) doi:10.1021/acs.jctc.1c00810.). If I compare the results from this manuscript to those of published Deep Docking studies, I don't see any major benefit here (speed or accuracy). How the authors would comment this aspect?*

The replacement of a docking calculation with a deep learning model is another approach to increase the efficiency

of a docking campaign. It is important to point out, however, that a model for docking will never be as accurate as a docking calculation; it approximates it. So while the deep network speeds things up, there is a cost to be paid in terms of accuracy. Our fragment docking approach avoids this compromise.

Reviewer #2

1. *Please check the grammar of the first sentence in the second paragraph of the introduction*

This sentence has been rewritten.

2. *Mentioning FlexNovo as the first approach in this context: what about combi-docking?*

A sentence (with citations) was added to include combi-docking.

3. *Please rephrase your differentiation to FlexNovo, as it is misleading in the current version: the FlexNovo space is restricted to splitting and recombining products ignoring reaction rules versus the fragment space for docking is restricted to validated reactions*

We have edited the comparison to FlexNovo to improve clarity.

4. *Please check: In the results chapter you describe 136,835 building blocks compared to 71,894 building blocks in the methods section*

A building block can give rise to more than one building block fragment (different reaction points on the building block). The first sentence of the Results section has been edited to add this clarification..

5. *You got 129,125 poses in the first step. Do you know, from how many building blocks? Or in other words, how many building blocks were declined by docking?*

The total number of building blocks is 136,835 (in the first sentence), and the best 500 were selected.

6. *In the first paragraph of the results chapter you selected the best 500 unique molecules. In my opinion, it is clearer to talk about the best 500 unique fragments or building blocks and the associated poses*

Agreed. We have changed the wording.

7. *You preselected the top scoring 33,000 virtual products. Is there any rationale for this number?*

The top 50,000 poses were chosen, and in cases where there was more than one pose for a molecule, the best pose was chosen, resulting in 33,000 distinct molecules. A sentence was added to clarify this.

8. *Please correct the typo in the first sentence of the second paragraph of the results chapter.*

The typo has been corrected.

9. *In the last paragraph of the results chapter I miss a reference or further details about compound 7. Obviously, it is not your hit no. 7 and where did you get the XRay structure from?*

This is a mistake. It is indeed one of our inhibitors, number 22. Thank you for catching this!

10. Supporting information: In the list of PDB codes are only 17 of 18 codes

3NDM was omitted from the list. This has been corrected.

11. Supporting information: which software and which Fingerprints did you use for the 2D similarity searches?

This information has been added to the Supporting Information

REVIEWER COMMENTS

Reviewer #1 (Remarks to the Author):

I have read the revised version of the manuscript and also the response letter. Based on these I have some comments, mainly related to some additional text and my original comments numbers (1, 3 and 4).

1. The authors have added quite a large analysis concerning the difference between individual X-ray structures and docking poses. I don't see any major benefit in this. The accuracy of different docking methods in producing the right docking pose is of course a critical study, but in this case the original text was surely enough.

2. As a response to my original comment number 1, the authors refer to the publication by Sadybekov et al and mention that "... they used the same ROCK1 pdb structure we used in our study. They compared their hit rate to a random selection of a full library and estimated a 40-fold enrichment. Our work corroborates theirs nicely and should give the scientific community confidence that the method is valid and useful."

This raises some new questions: I asked in my original comment the following "Although current results are good, it is still not clear if those results are comparable to the case where full library is used (so how many hits are missed)." The fact that authors refer to results of Sadybekov is not answering the question, because the authors approach should be evaluated, not the one by Sadybekov. I know that there are similarities in these approaches, but it do not guarantee same quality of results.

3. The authors mentioned in response to my original comment number 3 (effect of different docking methods) "The use of different scoring functions was a practical consideration. This effort was a collaboration, and one group used FlexX for docking (and the visualization that accompanies the HYDE scoring function), while the other used FRED for a docking engine. Interestingly, the work of Sadybekov, et al. used yet a different docking engine, and the results they got for the ROCK1 system are different from ours." On my mind this is conflicting. In previous point the authors state that Sadybekov results with ROCK1 do corroborates with the authors work. One should keep in mind that Sadybekov et al were using systematically the same approach in all of the docking. We all agree that every docking method has several issues. However, combining two (or more) methods do not mean that the issues will cancel each one out. It might result both positive or negative bias.

3. The response to original comment 4 "The replacement of a docking calculation with a deep learning model is another approach to increase the efficiency of a docking campaign. It is important to point out,

however, that a model for docking will never be as accurate as a docking calculation; it approximates it. So while the deep network speeds things up, there is a cost to be paid in terms of accuracy. Our fragment docking approach avoids this compromise." I must disagree here. It is absolutely true that machine learning is approximating docking results. However, the authors have not shown that the proposed method is giving same results as normal docking. Based on the method description it is "approximating" normal docking methods since initial selection of fragments is based on docking results of. The method proposed here is in this sense similar that the one of Yang et al, so the docking results are estimated by using fragment-based docking and normal docking is done only within a small subset of the original library. The method of Yang et al is doing the same as only small subset of compounds is actually docked but the final ranking (after model training and compound ranking) can be also corrected with (again) normal docking.

Response to Reviewers' Comments

Reviewer #1

1. *The authors have added quite a large analysis concerning the difference between individual X-ray structures and docking poses. I don't see any major benefit in this. The accuracy of different docking methods in producing the right docking pose is of course a critical study, but in this case the original text was surely enough.*

We defer to the editors on the level of detail in the analysis of the X-ray structures and docked poses and will edit the text accordingly.

2. *As a response to my original comment number 1, the authors refer to the publication by Sadybekov et al and mention that "... they used the same ROCK1 pdb structure we used in our study. They compared their hit rate to a random selection of a full library and estimated a 40-fold enrichment. Our work corroborates theirs nicely and should give the scientific community confidence that the method is valid and useful." This raises some new questions: I asked in my original comment the following "Although current results are good, it is still not clear if those results are comparable to the case where full library is used (so how many hits are missed)." The fact that authors refer to results of Sadybekov is not answering the question, because the authors approach should be evaluated, not the one by Sadybekov. I know that there are similarities in these approaches, but it do not guarantee same quality of results.*

We agree and have added a section in the Supplemental Information that compares results from fully enumerated docking to our approach. While the improvements in the docking hit rate using Chemical Space Docking (CSD) are demonstrated by the high enrichment, complete enumeration of the full library leads to hits that CSD missed. This emphasizes the importance of the initial fragment selection and is an important addition to the manuscript.

3. *The authors mentioned in response to my original comment number 3 (effect of different docking methods) "The use of different scoring functions was a practical consideration. This effort was a collaboration, and one group used FlexX for docking (and the visualization that accompanies the HYDE scoring function), while the other used FRED for a docking engine. Interestingly, the work of Sadybekov, et al. used yet a different docking engine, and the results they got for the ROCK1 system are different from ours." On my mind this is conflicting. In previous point the authors state that Sadybekov results with ROCK1 do corroborates with the authors work. One should keep in mind that Sadybekov et al were using systematically the same approach in all of the docking. We all agree that every docking method has several issues. However, combining two (or more) methods do not mean that the issues will cancel each one out. It might result both positive or negative bias.*

The most significant characteristic of CSD is the reduction in the search space done at the initial fragment stage. That is why we focused the new computational studies (added to the Supplemental Information) on that part of the process. FlexX (for pose generation) and HYDE (for scoring) were used alone at that stage, so the comparison between the CSD approach and the fully enumerated docking uses a single docking protocol. Once the best scoring compounds were obtained (~33k compounds down from ~1 billion), subsequent steps can be considered as filters: strain energy, clustering, and docking with a different program. We do not view this as combining docking methods, but rather adding a "second opinion" docking result as a filter. Consensus scoring approaches aimed at an improving robustness have been used before (e.g. https://doi.org/10.1007/978-981-15-8936-2_2).

4. *The response to original comment 4 "The replacement of a docking calculation with a deep learning model is another approach to increase the efficiency of a docking campaign. It is important to point out, however, that a model for docking will never be as accurate as a docking calculation; it approximates it. So while the deep network speeds things up, there is a cost to be paid in terms of accuracy. Our fragment docking approach*

avoids this compromise." I must disagree here. It is absolutely true that machine learning is approximating docking results. However, the authors have not shown that the proposed method is giving same results as normal docking. Based on the method description it is "approximating" normal docking methods since initial selection of fragments is based on docking results of. The method proposed here is in this sense similar that the one of Yang et al, so the docking results are estimated by using fragment-based docking and normal docking is done only within a small subset of the original library. The method of Yang et al is doing the same as only small subset of compounds is actually docked but the final ranking (after model training and compound ranking) can be also corrected with (again) normal docking.

We agree. Our method is not equivalent to full docking and can be viewed as model or approximation to full docking. We hope the reviewer agrees that the additional computational experiments sufficiently validate the approach.

REVIEWERS' COMMENTS

Reviewer #1 (Remarks to the Author):

The authors have responded to my comments quite well. Especially the new added text to supplementary material is fine. This additional test shows that the CSD approach is able to find majority of the good scorers. It is still very much an open question how well CSD actually works and in which cases it would give a good outcome. However, the current material in the manuscript allows interested reader to estimate if the method is good enough for her/his studies.

Response to Reviewers' Comments

Reviewer #1

The authors have responded to my comments quite well. Especially the new added text to supplementary material is fine. This additional test shows that the CSD approach is able to find majority of the good scorers. It is still very much an open question how well CSD actually works and in which cases it would give a good outcome. However, the current material in the manuscript allows interested reader to estimate if the method is good enough for her/his studies.

We are happy to hear that Reviewer #1 is satisfied with the additional tests and supplementary material. We are grateful for the suggestions that led to them and think the paper is stronger as a result.